# Characterization of Material Properties Based on Inverse Finite Element Modelling

**Mikdam Jamal [1],\* and Michael N. Morgan [2]**

[1]  The Manufacturing Technology Centre Ltd., Ansty Park, Coventry CV7 9JU, UK
[2]  Advanced Manufacturing Technology Research Laboratory (AMTREL), Faculty of Engineering
    and Technology (FET), Liverpool John Moores University, Liverpool L3 3AF, UK
\*  Correspondence: Mikdam.Jamal@the-mtc.org

**Abstract:** This paper describes a new approach that can be used to determine the mechanical properties of unknown materials and complex material systems. The approach uses inverse finite element modelling (FEM) accompanied with a designed algorithm to obtain the modulus of elasticity, yield stress and strain hardening material constants of an isotropic hardening material model, as well as the material constants of the Drucker–Prager material model (modulus of elasticity, cap yield stress and angle of friction). The algorithm automatically feeds the input material properties data to finite element software and automatically runs simulations to establish a convergence between the numerical loading–unloading curve and the target data obtained from continuous indentation tests using common indenter geometries. A further module was developed to optimise convergence using an inverse FEM analysis interfaced with a non-linear MATLAB algorithm. A sensitivity analysis determined that the dual spherical and Berkovich (S&B) approach delivered better results than other dual indentation methods such as Berkovich and Vickers (B&V) and Vickers and spherical (V&S). It was found that better convergence values can be achieved despite a large variation in the starting parameter values and/or material constitutive model and such behaviour reflects the uniqueness of the dual S&B indentation in predicting complex material systems. The study has shown that a robust optimization method based on a non-linear least-squares curve fitting function (LSQNONLIN) within MATLAB and ABAQUS can be used to accurately predict a unique set of elastic plastic material properties and Drucker–Prager material properties. This is of benefit to the scientific investigation of properties of new materials or obtaining the material properties at different locations of a part which may be not be similar because of manufacturing processes (e.g., different heating and cooling rates at different locations).

**Keywords:** material characterization; inverse finite element material modelling; elastic plastic material model; Drucker–Prager material model

## 1. Introduction

In this paper, outcomes of a research study concerned with the application of indentation processes with different indenter geometries are reported. The aim of the study was to establish a predictive capability for the elastic plastic material properties of various material systems and to develop an accurate method for specific applications. Many researchers have suggested that a non-unique set of mechanical properties can be predicted for strain hardening elastic plastic material from a single indentation test. However, most proposed methods in the literature have been characterised by parameters of load displacement curves using two or more indenters to determine a unique set of material properties [1–3].

Finite element modelling (FEM)-based algorithms using single and multiple indenters have been proposed by other researchers to determine the mechanical properties of different engineering material systems. This approach has been used on some complex or non-standard materials or surfaces, such as in vivo tension and brittle indentation [4–6]. Two different approaches were established to investigate and predict the elastic plastic material properties and other complex material constitutive laws. The first approach predicts material properties based on the single indentation test, examined separately, with all indenters sharing the same initial start value. The application of such an approach, which has been investigated by many researchers, has failed to achieve high accuracy as the range of tested material properties produced non-identical load displacement curves. In some cases the accuracy of this approach could be improved depending on the previous knowledge about the material [7–9].

The second approach predicts the material properties based on the dual indentation test. In this approach, two types of indenters with different shapes and dimensions are employed, resulting in different plastic strain profiles, i.e., different load displacement curves. Minsoo et al. [10] applied a dual triangular pyramidal indenters for material property evaluation. Chollacoop et al. [8] developed forward analysis, which considered that the representative stress and strain and the loading curvature were functions of the face angle of the conical indenter. The inverse algorithm then used the second pair of representative stress and strain values in order to obtain the unknown mechanical properties. The algorithm showed significant improvements in the predicted yield stress, $\sigma y$, and strain hardening exponent, 'n', compared to a single indenter.

Yan et al. [11] used dual indenter geometries to determine the mechanical properties in engineering materials. The modulus of elasticity and initial residual stress were assumed to be known. They performed forward FEM simulations with dual indenters to predict the yield stress and strain hardening. The result showed that the load displacement curves were more appropriate for the prediction of yield strength than the single indenter geometry approach, with an error of less than 5%.

The research in this study was focused on alternative approaches that use inverse FEM accompanied by an optimization algorithm to obtain and optimize the elastic plastic and Drucker–Prager material properties. It builds on the prior research of the authors reported in [12,13]. The proposed framework will enable the characterisation of complex material systems. The main objective of this research is to develop a coupled computational method based on FEM and an optimization algorithm to extract unique and accurate mechanical properties for an elastic plastic material model with isotropic hardening and a Drucker–Prager material model from full indentation loading–unloading curves using dual indenter geometries. The second objective was to examine the accuracy of the proposed inverse framework—coupled FEM with an optimization algorithm technique—based on available load-displacement data.

The work consists of two main parts. In the first part, inverse FEM models of the continuous indentation of commonly used 3-D indenter geometries (Vickers, Berkovich and spherical indenter) were developed. In the second part, an inverse framework—FEM interfaced with a non-linear MATLAB optimization algorithm—was developed, based on the load displacement results of dual indentation data. The effects of initially assigned values of the material properties and indenter geometry were examined to investigate the robustness of the proposed optimisation framework. The optimization framework was then used to predict material properties by matching the load-displacement curves.

## 2. Finite Element Indentation Models

Three-dimensional numerical models for three different axisymmetric rigid indenter geometries (Berkovich, Vickers, and spherical) were developed in ABAQUS to validate the optimization technique for various material systems. Figure 1a,b shows the 3-D Berkovich and Vickers indenter geometries—only 1:4 of symmetric specimens and indenters were performed for both indenters. The planes of symmetric geometries are defined in the X-Z and Y-Z planes. All specimens and indenters were modelled with eight-node element type reduced integration (C3D8R) and four-node element type rigid quadrilateral (R3D4), respectively. Both element types are used for stress and displacement

analysis. Figure 1c shows 3-D the spherical indenter where quarter of the specimen and indenter were modelled. Symmetry was defined on the X-Z and Y-Z planes. The specimen and indenter were modelled with eight-node reduced integration element (C3D8R) for the stress displacement analysis.

The indentation method was simulated in two alternating steps (loading and unloading). During the loading step, the indenter was moved along in the z-direction in ramp mode and penetrated the specimen until the maximum depth was achieved. During the unloading step, the indenter was returned to the initial position. The reaction force was recorded at the intender representing the total force during loading and unloading.

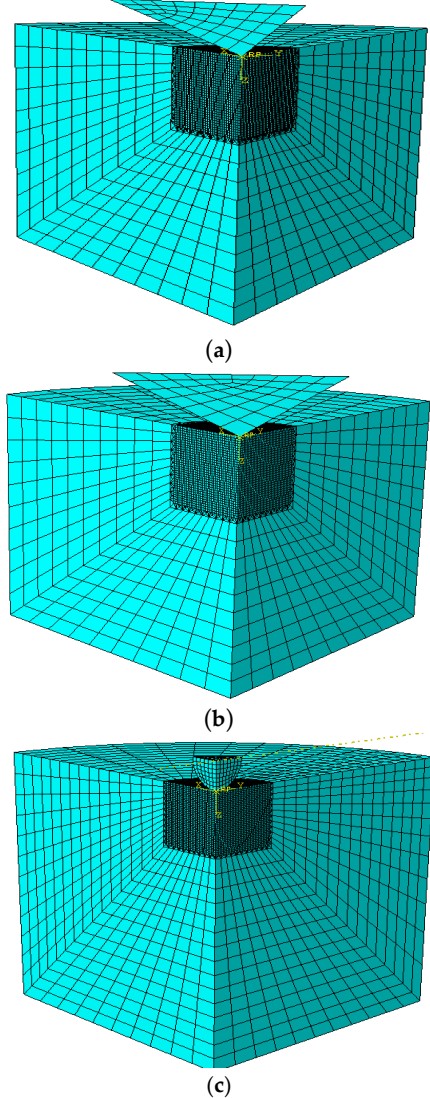

**Figure 1.** (**a**) FE model of Berkovich indentation. (**b**) FE model of Vickers indentation. (**c**) FE model of spherical indentation.

## 2.1. Elastic Plastic Material Constitutive Model

An elastic plastic material constitutive law was used in ABAQUS [14] to model the elastic and plastic behaviour of metallic alloys [15]. The constitutive law used to simulate the indentation process is shown in Equation (1). The elastic behaviour was modelled using Hooke's law while the plasticity was modelled using an isotropic strain hardening model described with a power function stress ($\sigma$) and strain ($\epsilon$).

$$\sigma = \begin{cases} E\varepsilon\,, for : \sigma \leq \sigma_y \\ R\varepsilon^n, for: \sigma \geq \sigma_y \end{cases}.$$ (1)

For modelling of true stress and true strain behaviour, [15] proposed Equation (2) to calculate the plasticity:

$$\sigma_p = R \left( \frac{\sigma_y}{E} + \varepsilon_p \right)^n.$$ (2)

The material coefficient, R, is given by:

$$R = E^n \sigma_y^{\,1-n}.$$ (3)

The four parameters $(E,\ \sigma_y, v, n\ )$ were used to optimize the elastic plastic material properties. The Poisson's ratio and strain hardening exponent values span the range between 0 and 0.5 for most engineering materials. However, in this study, no lower and upper boundaries have been specified for the values of the Young modulus and yield stress in order to represent a wide range of metallic and ceramic material properties. A fixed set of plastic strain values of $0 > \varepsilon_p \geq\ 0.3$ with 0.05 step increment were used. The stress values were then updated for a given plastic strain value using Equation (2).

### 2.2. Drucker–Prager Material Constitutive Model

Linear Drucker–Prager hardening constitutive material law was used to describe the indentation response of material exhibiting hydrostatic stress sensitivity behaviour. Such a model can be used to describe the deformation behaviour of soils and granular materials, metallic glass and polymer materials [16]. The linear plastic Drucker–Prager model is given by Equation (4):

$$\sigma_e\ + \mu\,\sigma_m - \sqrt{3}\,\sigma_S = 0\,,$$ (4)

where $\sigma_e = \sqrt{\frac{3}{2} S_{ij} S_{ij}}$ is the Von Mises equivalent stress, $S_{ij} = \sigma_{ij} - \sigma_m \varepsilon_{ij}$ is the stress deviator, $\sigma_m = \sigma_{kk}/3 = -p$, $\sigma_S$ is the shear stress, $\mu$ is the hydrostatic stress sensitivity parameter and $\beta$ is the friction angle. In ABAQUS, the Drucker–Prager material model is defined using the angle of friction $\beta$, dilatation angle $\psi$, flow stress ratio K and hardening curves for different strain rates.

The angle of friction β can be determined from the hydrostatic stress sensitivity parameter, while μ is dependent on the adhesive material and characterises the sensitivity of yielding to hydrostatic stress. The value of μ is determined from tests under two different stress states given by Equation (5), using yield stress from shear and tensile tests. The dilatation angle ψ can be determined from the flow parameter $\mu^*$ from Equation (6). Non-associated flow is defined when $\mu^*$ is not equal to μ, while associated flow is defined when $\mu^*$ is equal to μ. In this study, an associated flow was assumed by setting $\mu^*$ equal to $\mu$ in ABAQUS.

$$tan\,\beta = \mu = 3\big[\big(\sqrt{3}\,\sigma_s/\sigma_T\big) - 1\big]\,,$$ (5)

$$\mu^* = \tan\psi =\ 3(1 - 2\,v^p)/2(1 +\ v^p)\,,$$ (6)

where $v^p$ is a plastic component of Poisson's ratio.

The third parameter required in ABAQUS is the flow stress ratio K, which defines the differences in material behaviour under tension and compression. Park et al. [17] considered the parameter K to be in the range of $0.788\ \leq k \leq 1$ in order to ensure the convexity of the yield stress. However, the flow stress ratio used in this study for the material model was set to 1, assuming identical behaviour under tension and compression.

Three material parameters $(E,\ \sigma_{yc}, \beta\ )$ have been used to optimize the linear Drucker–Prager material model. In this study, no lower and upper limits have been specified for the values of the Young modulus and compressive yield stress. The friction angle values were selected to be in the range of $0o \geq \beta \leq 30°$, with 0.1° space interval in order to represent a wide range of material properties.

## 3. Development of the Optimization Method

Many optimization methods have been used by researchers to predict the best parameters using single- or multi-objective functions, for example, [18,19]. The main purpose of optimization techniques is the involvement of iteratively changing the material parameters by re-running the FEM until it achieves a best fit between the load displacement curve obtained from real measurement results and the curve obtained from numerical analysis. In this approach, an optimisation algorithm is coupled with the FEM in order to find the optimal values (minimum objective function) for a set for a wide range of material properties to be determined.

In this study, an optimization algorithm has been developed to determine the material properties for a given set of indentation data using an iterative procedure in MATLAB. The non-linear least-squares optimization function (LSQNONLIN) was developed in MATLAB based on the Levenberg–Marquardt algorithm (Matlab). A special code was written in MATLAB, including the optimisation function and commands to read input files, write output files and execute the ABAQUS solver. The optimisation process started by selecting arbitrary initial values for each parameter and then running the ABAQUS input file using these values for the particular material model. A python script was then used to extract the history of force and displacement which is read in MATLAB to compute the objective function.

The optimization algorithm based on the dual indentation method was also assessed to predict the elastic plastic material properties. In this case, the new optimization algorithm was developed to allow two sets of input data with different indenter types or size to be used. The MATLAB code was then used to automatically run two ABAQUS input files in order to iteratively determine the residual error between target and optimized load displacement curves. The residual error criterion is based on the use of objective function until minimum convergence value within the range of $0.001 \leq minF(x) \leq 0.02$ is achieved. The objective function for dual indenters is defined by Equation (7):

$$minF(x) = \frac{1}{2}\sum_{i=1}^{n}\left\{\left[\left(F_{num-l}^i - F_{exp-l}^i\right)^2 + \left(F_{num-ul}^i - F_{exp-ul}^i\right)^2\right]_{indenter1} + \left[\left(F_{num-l}^i - F_{exp-l}^i\right)^2 + \left(F_{num-ul}^i - F_{exp-ul}^i\right)^2\right]_{indenter2}\right\},\tag{7}$$

where $x = \left(E, v, \sigma_y, and\ n\right)$ for the elastic–plastic material law and $x = \left(E, v, \sigma_{yc}, and\ \beta\right)$ for the linear Drucker–Prager material law, $min\ F(x)$ is the minimum objective function and $x$ is the optimization parameter set. $F_{exp-l}^i$ is the measured force applied during loading at a particular depth, $F_{num-l}^i$ is the value of force obtained by the FEM at the same depth as in the experiment during loading, $F_{exp-ul}^i$ is the measured force in unloading at a particular depth, $F_{num-ul}^i$ is the value of force obtained by the FEM at the same depth as in the experiment during unloading and n is the number of sampling points in each test.

In this method, the objective function value was first calculated at each indentation point (displacement step) using a non-linear least-squares objective function (LSQNONLIN) in MATLAB, and then the sum of the objective functions were integrated over the whole indentation curve. The total objective function value for a given set of material parameters $(\sigma_y, E, v, n)$ was calculated by the summation the objective function of dual indenters at each iteration in the optimization algorithm. Figure 2 shows the optimization workflow for material characterization. In this workflow, the processing of input and output files were created, then the data extracted to the .rpt files in ABAQUS. The whole process was implemented into the automated algorithm for the final stage of inverse FEM analysis by a non-linear least-squares data fitting optimization tool.

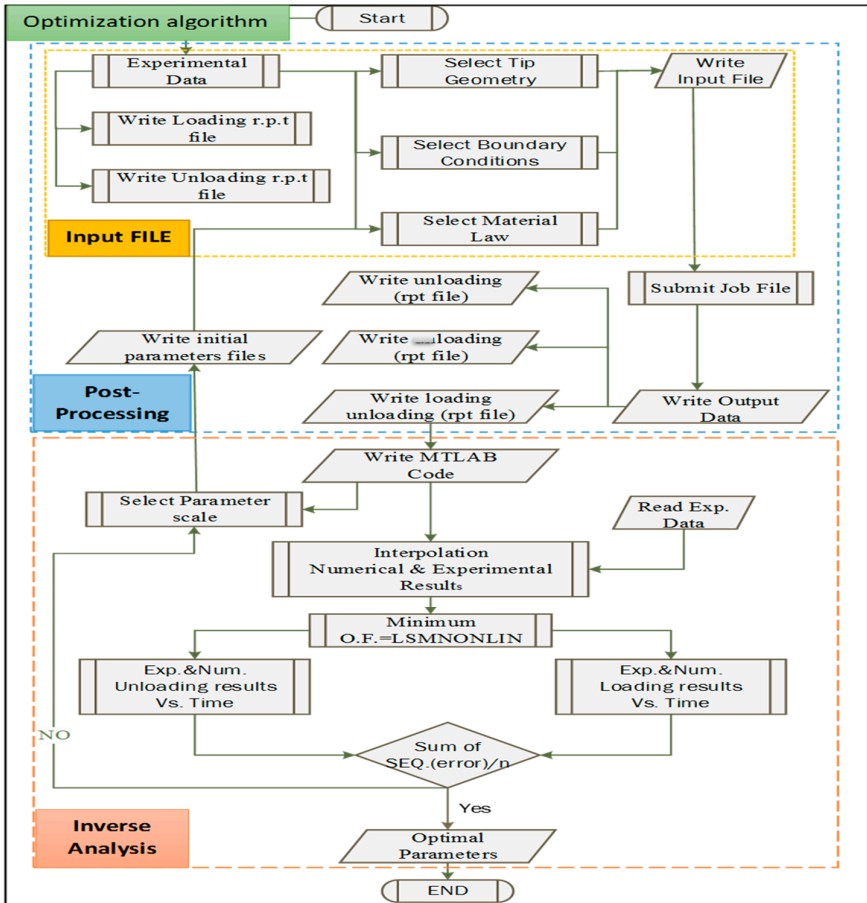

**Figure 2.** Optimization workflow for material characterization.

## 4. Results

### 4.1. Optimization Analysis of Elastic Plastic Material Properties Based on Dual Indenter Geometries

The optimization algorithm was carried out using 3-D indenter geometries (Berkovich, Vickers and spherical) to predict the material properties of an elastic plastic target material. Figure 3 shows the target numerical load displacement curve for pure aluminium material with known mechanical properties ($\sigma_y = 550MPa, E = 72GPa, v = 0.22, n = 0.1$) (Chollacoop et al. 2003 [10]), which is used as blind test numerical data based on the indentation process of three different tip geometries. Various ranges of initial guess were used to investigate the effect of starting point on the convergence of results. The numerical load displacement data were divided into 50 equally spaced points against the indentation force and used in the post-processing stage of the optimization workflow.

The numerical simulations of the target material show discrepancy in the loading–unloading curves for different indentation processes. These differences in the load displacement curves give a good boundary to test the sensitivity and accuracy of the optimization algorithm of elastic plastic materials. However, in order to validate the optimization algorithm in more depth, indentation hardness HIT—Equation (8), effective elastic modulus $E_{eff}$—Equation (9) and indentation depth ratio (final indentation depth to the maximum indentation depth) which represents the depth ratio of target material (hmax/hf) T divided by the depth ratio of optimized material/(hmax/hf) O—Equation (10), can be calculated from the optimal loading unloading curve using the Oliver and Pharr method and compared with results obtained from target loading unloading curves.
Indentation hardness:

$$H_{IT} = \frac{F}{A_P}.$$ (8)

Effective elastic modulus:

$$E_{eff} = \frac{\sqrt{\pi}}{2\beta} \frac{S}{\sqrt{A_P}}.$$ (9)

Depth ratio:

$$(h_{max}/(h_f)_t / (h_{max}/(h_f)_{opt}$$ (10). (10)

In this case, the sensitivity of this algorithm was examined by changing four material parameters ($\sigma_y, E, v,$ and $n$). Other parameters related to the specimen geometry and size, boundary conditions and applied load were fixed for all numerical simulations. The Young modulus, yield stress, Poisson's ratio and strain hardening values were selected within the range of $10 \leq E \leq 150$ GPa, $100$ MPa $\leq \sigma_y \leq 3$ GPa, $0.05 \leq v \leq 0.5$ and $0 \leq n \leq 0.5$, respectively. The optimization results are summarised in Table 1. The initial guess set was selected randomly from a range of material properties for various types of dual indenter numerical simulations. However, the percentage errors between the predicted results for a particular parameter and the target results for the same parameters can be calculated using the following expression:

$$Residual\ error\ \% = \left| \left[ 1 - \frac{predicted\ result - target\ result}{target\ result} \right] \times 100 \right| \%.$$ (11)

The initially-prescribed mechanical properties for an elastic plastic hardening material model ($\sigma_y, E, v, n$) were randomly changed in order to examine the sensitivity of this method. Table 1 summarises the optimization results of three different dual indenter geometries: Berkovich and Vickers (B&V), Vickers and spherical (V&S) and spherical and Berkovich (S&B). The initial guesses were selected from a wide range of material property sets for various dual numerical simulations.

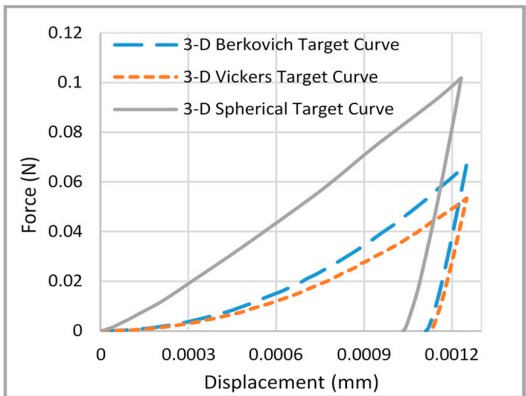

**Figure 3.** Target numerical load displacement curves determined from 3-D simulations for Berkovich indentation, Vickers indentation and spherical indentation.

The optimization analysis based on dual indenter geometries suggested that the four parameters ($\sigma_y, E, v, n$) achieved convergence at different iteration numbers to within 2% of the target values, regardless of the starting point. The result also shows that the objective function between the target and predicted load displacement curves was less than 1%. The optimized modulus of elasticity and strain hardening values are in excellent agreement with the target values. This suggests that the elastic plastic material properties can be accurately obtained by the proposed optimization technique of dual indenter geometries.

In order to examine the accuracy of the proposed method, Table 1 presents the calculation of the normalized hardness ratio HT/HO (target indentation hardness/optimized indentation hardness), and the normalized reduced modulus ratio (Er)T/(Er)O (target reduced modulus/optimized reduced

modulus). The results show that the maximum percentage error was about 1% in the reduced modulus and the hardness ratio over indentation techniques.

Figure 4 shows the convergence trends of the five initial guess values of elastic plastic materials. The results clearly illustrate that the initial guess values of elastic plastic hardening material models can converge to their target values by the dual indentation optimization algorithm, but with different iteration numbers. It is worth noting that additional analyses were also investigated using a wide range of initial guess values. It was found that the application of the proposed algorithm was more reliable for any initial guess values within the defined database, i.e., $(1 \leq E \leq 220)$ GPa, $100$ MPa $\leq \sigma_y \leq 3$ GPa, $0 \leq n \leq 6, 0.05 \leq v \leq 0.5$.

**Table 1.** Dual indenter optimization results of elastic plastic material. HT/HO = target indentation hardness/optimized indentation hardness, (Er)T/(Er)O = target reduced modulus/optimized reduced modulus.

| Indenter | Parameter | Target Value | Initial Value | Predicted Value | Error % | HT/HO | (Er)T/(Er)O | Depth Ratio |
|---|---|---|---|---|---|---|---|---|
| (3-D) Berkovich and Vickers | E(GPa) | 72 | 10 | 72.71 | 0.99 | 0.992 | 0.996 | 0.98 |
| | $\sigma_y$(MPa) | 550 | 260 | 540 | 1.85 | | | |
| | v | 0.22 | 0.14 | 0.222 | 1.2 | | | |
| | n | 0.1 | 0.01 | 0.099 | 0.99 | | | |
| (3-D) Vickers and spherical | E(GPa) | 72 | 90 | 72.43 | 0.59 | 0.997 | 0.992 | 0.978 |
| | $\sigma_y$(MPa) | 550 | 260 | 543 | 1.28 | | | |
| | v | 0.22 | 0.35 | 0.222 | 1.25 | | | |
| | n | 0.1 | 0.05 | 0.099 | 0.87 | | | |
| (3-D) Spherical and Berkovich | E(GPa) | 72 | 50 | 72.22 | 0.30 | 0.990 | 0.994 | 0.986 |
| | $\sigma_y$(MPa) | 550 | 440 | 546 | 0.47 | | | |
| | v | 0.22 | 0.2 | 0.222 | 1.11 | | | |
| | n | 0.1 | 0.01 | 0.101 | 1.01 | | | |

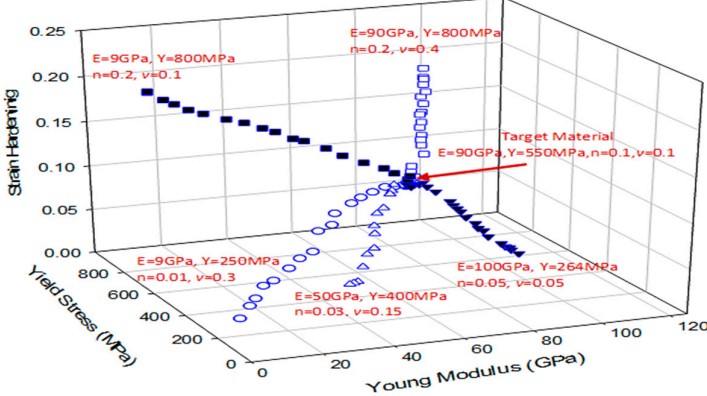

(a)Elastic plastic hardening target material

**Figure 4.** Converging trends of five initial guess values using spherical and Berkovich (S&B) dual indenter for (**a**) elastic plastic hardening target material.

Figure 5 shows the optimization history of the material properties from initial guess values to their target values (with 0.01 residual error) based on three different dual indentation tests (B&V), (V&S) and (S&B). The average convergence history of the indentation tests shows that the four parameters achieved the target values after 19, 17 and 14 iterations, respectively, over a range of initial guess material properties. The error bar presented in each column explains that material properties can reach their target values at different numbers of iterations, these variations depending on initial guess values.

The optimization process, based on the S&B indentation test, provides the best solution, as fewer iterations are required for the main parameters ($\sigma_y, E, n$) to achieve convergence. It can be clearly noticed that the Poisson's ratio required less iterations to achieve convergence;, whereas the Young modulus required a high number of iterations to achieve convergence, followed by the yield stress and then the strain hardening.

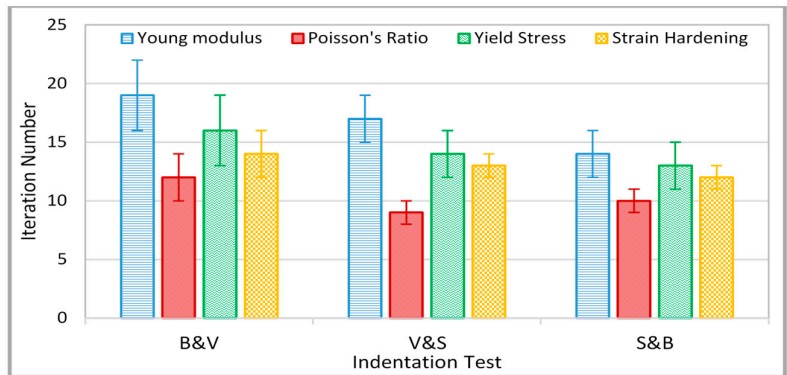

**Figure 5.** Optimization results of elastic plastic material properties based on (B&V), (V&S) and (S&B) indentation tests.

### 4.1.1. Sensitivity Analysis of Elastic Plastic Optimization Algorithms

The sensitivity of the optimization process used to predict elastic plastic material properties as a result of continuously changing the input parameters until achieving a best match between predicted and experimental is a major difficulty in using the inverse or reverse method [20]. In this study, a series of input target materials were employed to investigate the sensitivity and accuracy of the optimization algorithm based on S&B, B&V and V&S indentation methods. However, in the actual experimental work, there are many factors which can potentially cause systematic and random error. These errors may be related to indenter deformation and tip blunting during indentation, as well as the accuracy of the indentation measurements [8].

Figure 6 shows the sensitivity analysis of three optimization methods with five different sets of material properties which have been used as input data to evaluate the accuracy and sensitivity of the approaches. In each approach, there are only a few material property sets that match the target data, and all parameters are focused in a small boundary region. As displayed, the results achieved by the S&B approach is significantly better than the other methods (B&V and V&S) because the boundary regions are smaller. A small deviation in the predicted mechanical properties ($\sigma_y, E, n$) produces a very limited material range with identical load displacement curves (same objective function); such behaviour reflects the uniqueness of the method in solving complex material systems.

Table 2 summarizes the sensitivity analysis of the S&B optimization method applied on five different set of material properties using theoretical values. The results from each set of parameters represent the residual error between the target and predicted load displacement curves to within ≤2% determined by the non-linear least-squares objective function (LSQNONLIN) in MATLAB. The previous analysis shows that the Poisson's ratio had less influence on the predicted load displacement curves, therefore, only three parameters ($\sigma_y, E, n$) were used in the optimization algorithm.

In the case of the S&B approach, the deviation and percentage error of E calculated during the sensitivity analyses for a range of materials were within 2.6 GPa and 1.6%, respectively. The deviation and percentage error of $\sigma_y$ were within 6.5 MPa and 1.1%, respectively, while the percentage error of n was within 0.001 and 1.2%, respectively. This suggests that the elastic modulus, yield stress and strain hardening can be extracted using the proposed method within 1.6%, 1.1% and 1.2% relative error, respectively. All the proposed parameters can be determined with a specific percentage of errors if the load displacement curves are measured with accuracies greater than 98%. This indicates that the accuracy of the measured load displacement curve is important to predict accurate material

properties. However, the results achieved are significantly better than some stated methods in previous works [8].

The true stress–strain curves with optimal predicted material properties (minimum objective function) are plotted in Figure 7, which shows that these stress–strain curves are identical. Figure 8 compares the load displacement curves of optimal material property sets with the input target data ($\sigma_y = 550\ MPa, E = 80GPa, v = 0.2, n = 0.09$).

The load displacement curves of the predicted material properties agree very well with the target material, all parameters being focused in a small boundary region to within ≤2% residual error. This suggests the optimization algorithm based on the pair of spherical and Berkovich indentations can accurately predict the elastic plastic material properties with unique stress–strain curves.

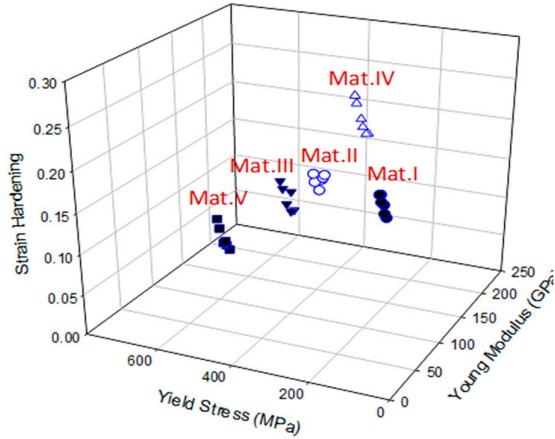

*B&V indentation method*

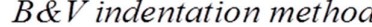

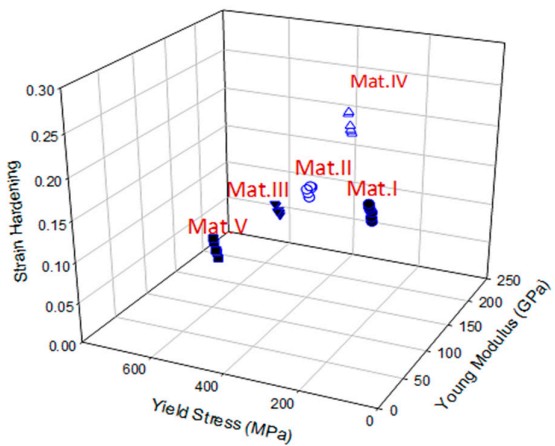

*S&B indentation method*

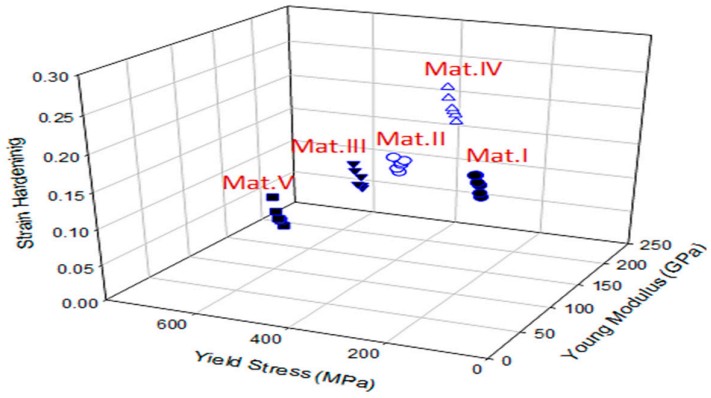

**Figure 6.** Sensitivity and accuracy results of elastic plastic optimization algorithms (S&B, B&V and V&S).

**Table 2.** Sensitivity and accuracy analysis of the S&B optimization method.

| Material | Parameter | Theoretical Value | Initial Value | Predicted Value | Error % |
|---|---|---|---|---|---|
| Mat.I | E(GPa) | 100 | 40 | 100.9 | 0.82 |
| | $\sigma_y$(MPa) | 160 | 100 | 161.2 | 0.71 |
| | v | 0.2 | 0.2 | 0.2 | 0 |
| | n | 0.16 | 0.1 | 0.16 | 0.99 |
| Mat.II | E(GPa) | 120 | 60 | 121.4 | 1.1 |
| | $\sigma_y$(MPa) | 350 | 175 | 346.6 | 0.98 |
| | v | 0.2 | 0.2 | 0.2 | 0 |
| | n | 0.15 | 0.05 | 0.15 | 0.87 |
| Mat.III | E(GPa) | 160 | 70 | 157.4 | 1.6 1.1 |
| | $\sigma_y$(MPa) | 500 | 250 | 544.5 | 0 |
| | v | 0.2 | 0.2 | 0.2 | 1.11 |
| | n | 0.1 | 0.01 | 0.09 | 1.01 |
| Mat.IV | E(GPa) | 200 | 110 | 202.4 | 1.2 |
| | $\sigma_y$(MPa) | 350 | 150 | 353.4 | 0.98 |
| | v | 0.2 | 0.2 | 0.2 | 0 |
| | n | 0.2 | 0.05 | 0.202 | 1.2 |
| Mat.V | E(GPa) | 80 | 10 | 79.2 | 1.0 |
| | $\sigma_y$(MPa) | 550 | 100 | 556.5 | 1.1 |
| | v | 0.2 | 0.2 | 0.2 | 0 |
| | n | 0.09 | 0.01 | 0.089 | 0.99 |

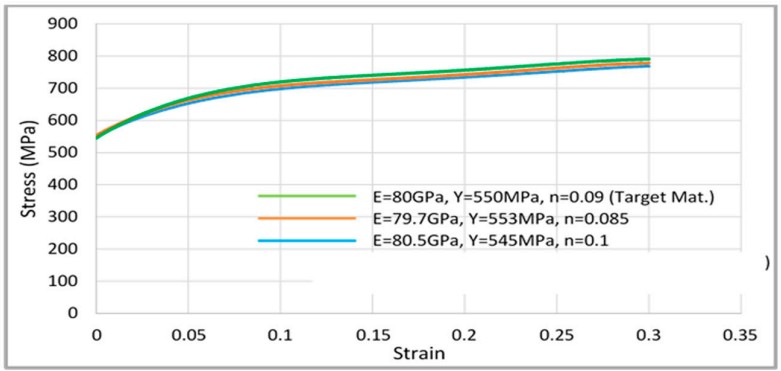

**Figure 7.** True stress-strain curves of target material and other data with objective function $\leq 0.01$.

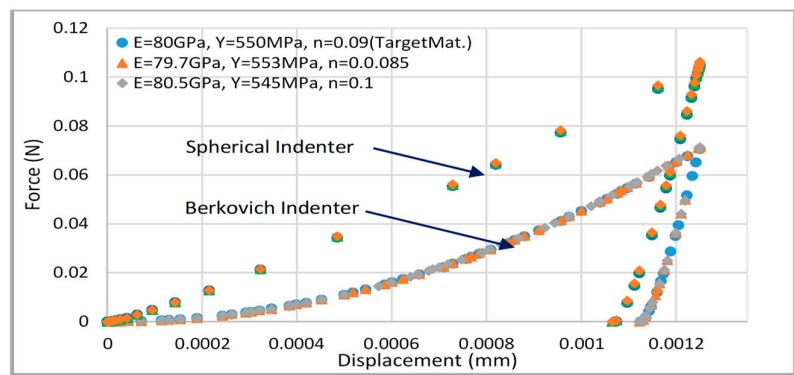

**Figure 8.** Comparison between the predicted load displacement curves using the optimal and target material properties for the S&B approach.

### 4.2. Optimization Analysis of Drucker–Prager Material Properties Based on Dual Indenter Geometries

The optimization algorithms based on the dual indentation method were also developed to predict the linear Drucker–Prager material properties. The optimization algorithms were carried out using the same procedure and principles used in the dual indenter geometries for elastic plastic materials. Combinations of 3-D indenter geometries (Berkovich, Vickers and spherical) were performed to predict the Drucker–Prager material behaviour. Various ranges of initial guess were used to investigate the accuracy and sensitivity analysis of the proposed approaches. Figure 9 shows the target load displacement curves for bulk metallic glasses (BMG) obtained numerically with known mechanical properties ($\sigma_{yc} = 1640\ MPa, E = 72 GP, v = 0.22, \beta = 30, \psi = 30,\ k = 1$) [21].

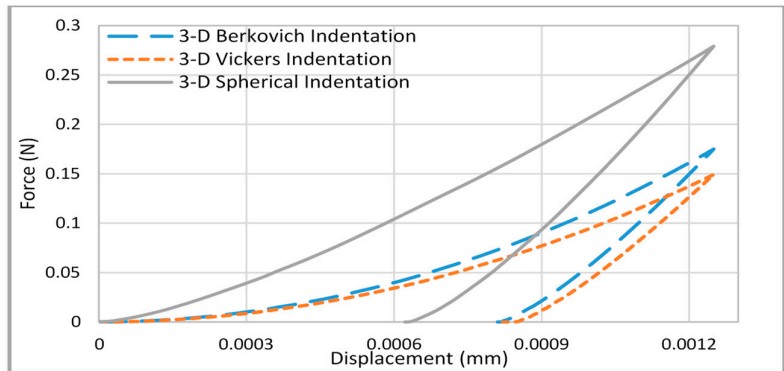

**Figure 9.** Target Drucker–Prager numerical load displacement curves determined from 3-D finite element modelling (FEM) simulations for (**a**) Berkovich, (**b**) Vickers and (**c**) spherical indentations.

The optimization processes include three different pairs of indenter geometries (B&V), (V&S) and (S&B). The initial guess mechanical properties of Linear Drucker–Prager hardening material ($E$, $\sigma_{yc}$, $\beta$) were changed a number of times in each process in order to investigate the sensitivity of this method. Table 3 summarises the optimization results of three different indentation tests based on the dual indentation methods on the BMG material.

**Table 3.** Dual indenter optimization results of hydrostatic stress sensitive (bulk metallic glasses, BMG) material.

| Indenter | Parameter | Target Value | Initial Value | Predicted Value | Error % | HT/H O | (Er)T/(Er) O | Depth Ratio |
|---|---|---|---|---|---|---|---|---|
| 3-D | E(GPa) | 72 | 50 | 73.39 | 1.89 | | | |
| (B&V) | $\sigma_{yc}$(GPa) | 1.64 | 1 | 1.61 | 1.82 | 0.97 | 0.96 | 0.97 |
| | β 0 | 300 | 150 | 30.4 | 1.31 | | | |
| 3-D | E(GPa) | 72 | 90 | 73.26 | 1.72 | 0.98 | 0.97 | 0.98 |
| (V&S) | $\sigma_{yc}$(GPa) | 1.64 | 1 | 1.62 | 1.23 | | | |

| | | | | | | | | |
|---|---|---|---|---|---|---|---|---|
| | β 0 | 300 | 400 | 29.8 | 0.6 | | | |
| 3-D | E(GPa) | 72 | 10 | 72.94 | 1.28 | | | |
| (S&B) | $\sigma_{yc}$(GPa) | 1.64 | 1 | 1.62 | 1.23 | 0.985 | 0.99 | 0.984 |
| | β 0 | 300 | 0.050 | 30.2 | 0.7 | | | |

The initial guess material properties were selected from a range of material property sets for various dual numerical simulations. The optimization algorithms were carried out by automatically changing the material properties in the ABAQUS input file (.inp) of each iteration until the objective function between the target and predicted load displacement curves achieved the minimum convergence value within the range of $0.001 \leq minF(x) \leq 0.02$. Despite using a range of initial guess parameters, the variables ($E$, $\sigma_{yc}$, $\beta$) can converge to their target values at different iteration numbers to within a 2% percentage error. The optimized reduced modulus and hardness ratio are in good agreement with the target values. This suggests that the linear Drucker–Prager material properties can be accurately obtained by the proposed optimization techniques of dual indenter geometries.

Figure 10 shows the convergence trends of five initial guess values of hydrostatic stress-sensitive plastic to their target material using the S&B indentation technique. The results demonstrated that the initial guess values could converge to their target values by the dual indentation optimization algorithm with different numbers of iterations. The materials with less difference between the initial and target values (i.e., availability of prior knowledge) will require fewer iterations to achieve convergence. Additional analyses were also investigated using a wide range of initial guess values. It was found that the application of the proposed algorithm is more reliable for any initial guess values within the defined database, i.e., 1 GPa $\leq$ E $\leq$ 150 GPa, 100 MPa $\leq \sigma_y \leq$ 5 GPa, $0^0 \leq \beta \leq 30^0$ and $0.05 \leq v \leq 0.5$.

It should be noted that the most important challenge of such an optimization algorithm is to identify the accuracy of the final predicted material property values based on real experimental tests for new materials where target values may be unknown. However, the solution of repeating the process several times with different initial guess values can overcome this problem and ensure the repeatability of numerical simulations.

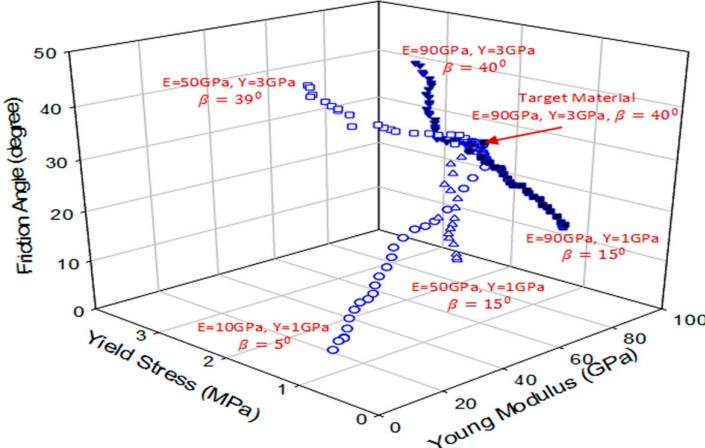

**Figure 10.** Converging trends of five initial guess values using S&B dual indenter for pressure-sensitive plastic material properties.

Figure 11 shows a comparison between three dual indentation methods concerning the optimization history of the initial guess material properties to their target values. The average convergence history of the B&V, V&S and S&B indentation tests shows that the three parameters achieved their target values after 49, 45 and 38 iterations, respectively, over a range of initial guess material properties. The error bar presented in each column explains that the material properties can reach their target values at different iteration numbers depending on initial guess values. The

optimization process based on the S&B indentation test provide the best solution, as fewer iterations are required for the main parameters ($E$, $\sigma_{yc}$, $\beta$) to achieve convergence.

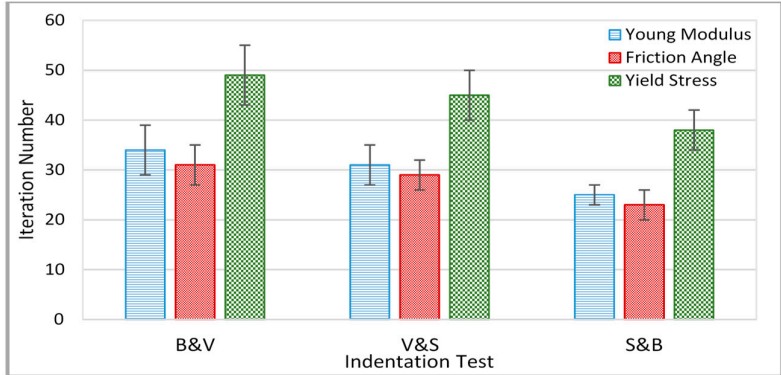

**Figure 11.** Optimization history of ($E$, $\sigma_{yc}$, $\beta$) based on B&V, V&S, and S&B indentation tests.

### 4.2.1. Sensitivity Analysis of Drucker–Prager Optimization Algorithms

A series of FEM simulations were developed to examine the accuracy and sensitivity of the optimization algorithms based on S&B, B&V and V&S indentation methods using a range of hydrostatic stress sensitive plastic material properties. Table 4 presents the material properties of bulk metallic glass BMG material used in the numerical simulations. The material sets were employed as an input target data (blind test data) to evaluate the accuracy and sensitivity of the methods.

Figure 12 shows the sensitivity analysis of three optimization methods with four different sets of BMG material properties presented in Table 4. As presented, there were few material property sets that matched the target data with the minimum objective function, and all parameters were concentrated in a small boundary region. The residual errors between target values and optimized parameters were varied according to the optimization algorithm type; however, the results achieved by the S&B method were significantly better compared with the other methods. The maximum relative errors were estimated as 7.5%, 6% and 3.5% in the B&V, V&S and S&B tests, respectively. Consequently, the predicted properties ($E$, $\sigma_{yc}$, $\beta$) produce a very limited material range, having identical load displacement curves (same objective function); such behaviour reflects the uniqueness of the method in solving complex material systems. However, the satisfactory existence of uniqueness and stability can suggest of considering the proposed method as a well-posed optimization solution.

**Table 4.** Bulk metallic glass material properties.

| Material | E (GPa) | $\sigma_{yc}$ (GPa) | $\beta$ (degree) | v | k | $\psi$ (degree) | References |
|----------|---------|---------------------|------------------|------|---|-----------------|------------|
| BMG I | 124 | 2.01 | 40 | 0.25 | 1 | 40 | (Inoue et al. 2001) |
| BMG II | 92 | 1.34 | 36 | 0.23 | 1 | 36 | (Saida et al. 2007) |
| BMG III | 74 | 1.78 | 35 | 0.23 | 1 | 35 | (Yuan et al. 2003) |
| BMG IV | 52 | 1.08 | 32 | 0.22 | 1 | 30 | (Inoue et al. 1989) |

In the case of the S&B optimization algorithm, the convergence of the elastic modulus E, yield stress $\sigma_{yc}$ and friction angle $\beta$ for the examined materials was within 4%, 3.65% and 4.2%, respectively. This demonstrated that the material properties $E$, $\sigma_{yc}$ and $\beta$ can be extracted using the proposed method to within 4%, 3.65% and 4.2% relative error, respectively. All the proposed parameters can be determined with a specific percentage of errors if the load displacement curves are measured with accuracies greater than 97%.

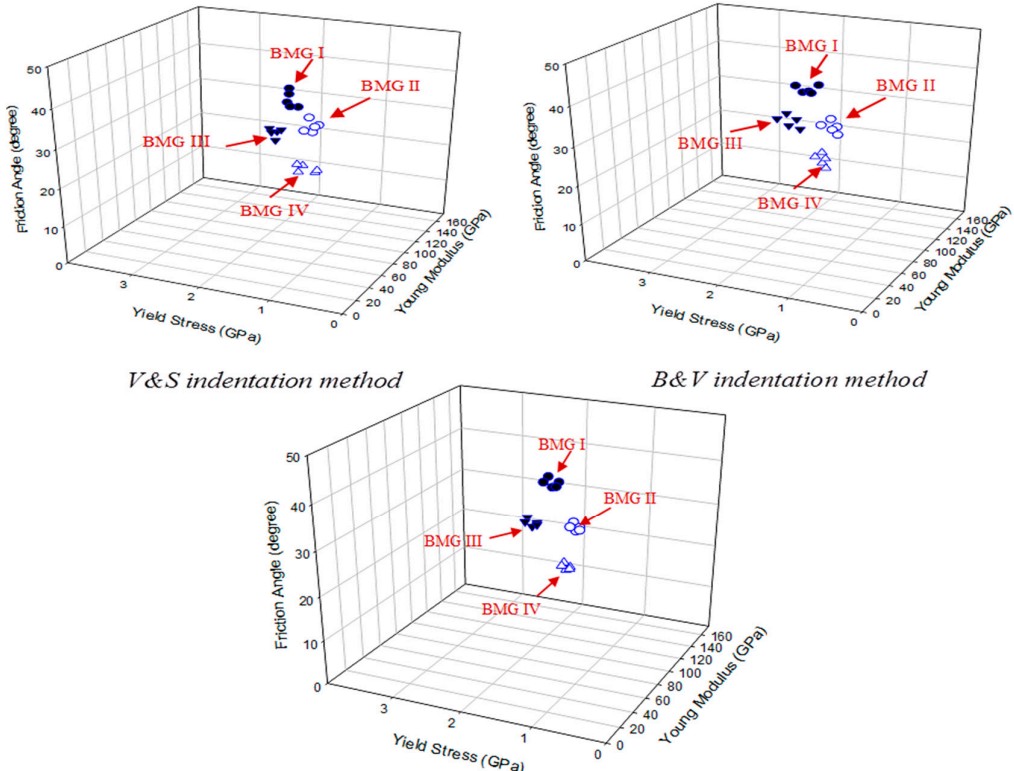

**Figure 12.** Illustration that there were a few material data sets over a small region matching the input data. This suggests that the combination of spherical and Berkovich indenters could produce unique results.

Figure 13 shows the sensitivity analysis of the dual indentation S&B optimization algorithm was expanded to include other material systems, such as ceramics, polymers, concrete and BMG. Table 5 summarizes the several material properties ($E$, $\sigma_{yc}$, $\beta$) used as input data to numerical simulations. The relative error for each parameter was measured at the best match between the predicted and target load displacement curves to within an accuracy of less than 3% determined by the non-linear least-squares objective function LSQNONLIN.

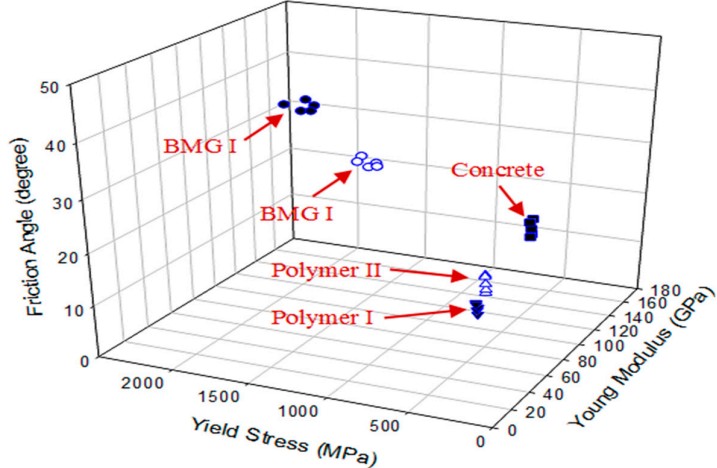

**Figure 13.** Sensitivity and accuracy results of S&B optimization algorithms.

**Table 5.** Sensitivity and accuracy analysis of S&B optimization method.

| Material | Parameter | Target Value | Initial Value | Predicted Value | Error % |
|---|---|---|---|---|---|
| Polymer I | $E(GPa)$ | 5 | 0.5 | 5.18 | 3.4 |
| (Seltzer et al. 2011) | $\sigma_{yc}(MPa)$ | 140 | 10 | 144.5 | 3.2 |
| | $\beta^0$ | $20^0$ | 0.050 | $22^0$ | 9 |
| Polymer II | $E(GPa)$ | 3.5 | 0.25 | 3.65 | 4.2 |
| (Seltzer et al. 2011) | $\sigma_{yc}(MPa)$ | 90 | 10 | 92.61 | 2.9 |
| | $\beta^0$ | $25^0$ | 20 | $23^0$ | 8.6 |
| Concrete | $E(GPa)$ | 40 | 3 | 38.2 | 4.5 |
| (Mokhatar and | $\sigma_{yc}(MPa)$ | 40 | 5 | 41.56 | 3.9 |
| Abdullah 2012) | $\beta^0$ | $30^0$ | 50 | $31.5^0$ | 5 |
| BMG I | $E(GPa)$ | 124 | 10 | 127.72 | 3 |
| (Inoue et al. 2001) | $\sigma_{yc}(MPa)$ | 2010 | 300 | 2076 | 3.3 |
| | $\beta^0$ | $35^0$ | 10 | $32.9^0$ | 4.7 |
| BMG II | $E(GPa)$ | 92 | 5 | 89.5 | 2.7 |
| (Saida et al. 2007) | $\sigma_{yc}(MPa)$ | 1340 | 200 | 1380 | 3 |
| | $\beta^0$ | $30^0$ | 10 | $28.65^0$ | 4.5 |

Figure 14 shows a comparison of load displacement curves between the predicted and input target data ($\sigma_y = 1.76\ GPa, E = 74\ GPa, v = 0.22, \beta = 32^0$). It is clearly demonstrated that the load displacement curves using the predicted material properties agreed very well with the input numerical target data. This suggests that the optimization algorithm based on dual of spherical and Berkovich indentations can accurately predict the hydrostatic stress-sensitive plastic material properties.

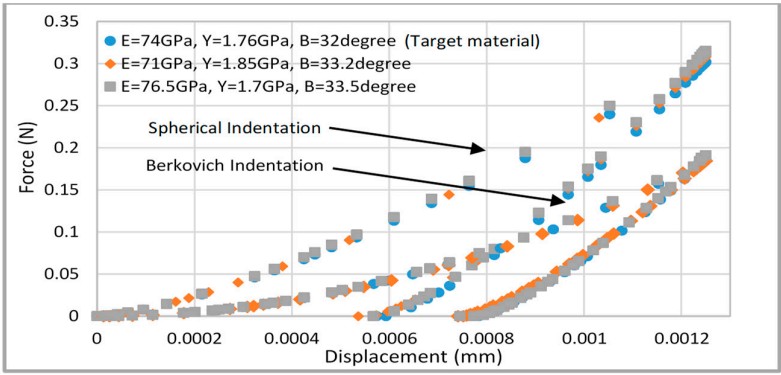

**Figure 14.** Comparison of load displacement curves between the predicted and target pressure-sensitive plastic material properties using the S&B approach.

## 5. Conclusions

In this study, an optimization algorithm was developed to extract the mechanical properties for a given set of indentation data using a non-linear least-squares curve fitting function (LSQNONLIN) within the optimization toolbox of MATLAB, based on the Levenberg–Marquardt algorithm. A special code was written in MATLAB, including an optimization algorithm as well as functions to read ABAQUS input files, write results files and execute ABAQUS. The optimization process started by selecting arbitrary initial values for the mechanical properties and then running ABAQUS models. A python script was then used to extract the history of load displacement data which was used to compute the objective function. The process runs iteratively until the best fit is achieved between predicted and experimental load displacement curves—this is achieved when the objective function reaches its minimum set by the convergence criteria. The optimum values of the parameters are selected when best fit between numerical and experimental or target data is achieved.

The dual indentation optimization process was established to predict the mechanical properties over a wide range of material constitutive laws (the elastic plastic material model and Drucker–Prager material model) to investigate the effectiveness of the optimization techniques for a wide range of materials. The results also show that the elastic modulus and yield stress require more iterations to reach convergence compared with other parameters. The optimization history of the full set of material properties for different indentation techniques clearly demonstrates that the dual indentation method delivers better convergence values despite a large variation in the starting parameter values and/or material constitutive model.

In this case, the S&B dual indentation approach and different initial guess material property values were also used to investigate the robustness of the proposed optimization algorithm. The results shows that an accurate Young modulus, yield stress and strain hardening were obtained and compared with the traditional technique of the Oliver and Pharr method based on experimental load displacement curve analysis. This is of benefit to the scientific investigation of properties of new materials.

**Author Contributions:** Formal analysis, M.J. and M.M.; Investigation, M.J. and M.M.; Methodology, M.J.; Software, M.J.; Supervision, M.J. and M.M.; Validation, M.J.; Writing – review & editing, M.J. and M.M.

**Conflicts of Interest**: The authors declare no conflict of interest.

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
