# Peer review of "Characterization of Material Properties Based on Inverse Finite Element Modelling"

_inventions, doi:10.3390/inventions4030040_

Round 1

Reviewer 1 Report

The topic is interesting, and the manuscript includes valuable results. There are a few major comments listed below: 1) English language requires editing since, although the paper is well written, it still holds many flaws and some sentences are a bit long. 2) The novelty of the manuscript is unclear. What is the main difference between the present work and published papers on the same topic? It needs to be clarified in the introduction section. 3) Literature review needs to be updated. Some of new references on anisotropic material characterisation can be found below:https://iopscience.iop.org/article/10.1088/0964-1726/25/10/105034/meta4) More details on the finite element method (see the following ref.) and inelasticity implementation should be provided.https://www.sciencedirect.com/science/article/pii/S02638223140041765) How did you implement the contact analysis into the FEM?

Author Response

Reviewer 1. 1) English language requires editing since, although the paper is well written, it still holds many flaws and some sentences are a bit long. Author Response: The paper has been modified as requested. 2) The novelty of the manuscript is unclear. What is the main difference between the present work and published papers on the same topic? It needs to be clarified in the introduction section. Author Response: The novelty is set out from line 70+. In the opinion of the author’s the statements are sufficiently strong. 3) Literature review needs to be updated. Some of new references on anisotropic material characterisation can be found below:https://iopscience.iop.org/article/10.1088/0964-1726/25/10/105034/meta4) More details on the finite element method (see the following ref.) and inelasticity implementation should be provided.https://www.sciencedirect.com/science/article/pii/S02638223140041765) How did you implement the contact analysis into the FEM? Author Response: Further articles have been referenced.

Reviewer 2 Report

In this work the authors proposed an inverse FE model to obtain elastic modulus, yield stress and so on of isotropic materials. The experimental procedure and the results were well adressed and discussed, they are interesting and useful for future development and applications.
This referee has not any suggested revisions under the techincal and theroetical point of view and no errors have been found.
The publication on inventions journal is suggested after the following (very) minor revisions:
1. The abstract is too long. It is suggested to reduce its extension.
2. Check Fig.4 and its caption, please. There is only one image in this figure, there is not need to write the case (‘a’) in the image as well as in the caption;
3. In Fig. 5, avoid the use of text in the figure. Use letters a, b and c to identify the different  plots and explain them in the figure’s caption.
4. Check typos in the conclusion section.

Author Response

Reviewer 2. This referee has not any suggested revisions under the technical and theoretical point of view and no errors have been found.